# The Effect of Antibiotics Treatment on the Maternal Immune Response and Gut Microbiome in Pregnant and Non-Pregnant Mice

**DOI:** 10.3390/nu15122723

**Published:** 2023-06-12

**Authors:** Marijke M. Faas, Yuanrui Liu, Lieske Wekema, Gisela A. Weiss, Carolien A. van Loo-Bouwman, Luis Silva Lagos

**Affiliations:** 1Immunoendocrinology, Division of Medical Biology, Department of Pathology and Medical Biology, University Medical Center Groningen, Hanzeplein 1, 9713 GZ, Groningen, The Netherlands; y_liu01@outlook.com (Y.L.); l.wekema@umcg.nl (L.W.); l.a.silva.lagos@umcg.nl (L.S.L.); 2Department of Obstetrics and Gynaecology, University Medical Center Groningen, Hanzeplein 1, 9713 GZ Groningen, The Netherlands; 3Yili Innovation Center Europe, Bronland 12 E-1, 6708 WH Wageningen, The Netherlands; adrienne.weiss@yili-innovation.com (G.A.W.); carolien.vanloo@yili-innovation.com (C.A.v.L.-B.)

**Keywords:** pregnancy, gut microbiota, immune response

## Abstract

The gut microbiota are involved in adaptations of the maternal immune response to pregnancy. We therefore hypothesized that inducing gut dysbiosis during pregnancy alters the maternal immune response. Thus, pregnant mice received antibiotics from day 9 to day 16 to disturb the maternal gut microbiome. Feces were collected before, during and after antibiotic treatment, and microbiota were measured using 16S RNA sequencing. Mice were sacrificed at day 18 of pregnancy and intestinal (Peyer’s patches (PP) and mesenteric lymph nodes (MLN)) and peripheral immune responses (blood and spleen) were measured using flow cytometry. Antibiotic treatment decreased fetal and placental weight. The bacterial count and the Shannon index were significantly decreased (Friedman, followed by Dunn’s test, *p* < 0.05) and the bacterial genera abundance was significantly changed (Permanova, *p* < 0.05) following antibiotics treatment as compared with before treatment. Splenic Th1 cells and activated blood monocytes were increased, while Th2, Th17 and FoxP3/RoRgT double-positive cells in the PP and MLNs were decreased in pregnant antibiotics-treated mice as compared with untreated pregnant mice. In addition, intestinal dendritic cell subsets were affected by antibiotics. Correlation of immune cells with bacterial genera showed various correlations between immune cells in the PP, MLN and peripheral circulation (blood and spleen). We conclude the disturbed gut microbiota after antibiotics treatment disturbed the maternal immune response. This disturbed maternal immune response may affect fetal and placental weight.

## 1. Introduction

During pregnancy, many alterations in the maternal immune response are observed [1]. This is necessary to protect the semi-allogeneic fetus from rejection [1]. Various changes are observed in the peripheral immune response, such as a decreased Th1/Th2 ratio and decreased Th17 cells, an increase in regulatory T (Treg) cells and an increase in the numbers and activational status of innate immune cells, such as monocytes and granulocytes [2,3,4]. This activational status of the innate immune system is also associated with a change in the monocyte subsets [5]. At the maternal–fetal interface, there is an increase in NK cells and macrophages [6]. These cells are important for the development of the placenta and the fetus [7].

The study of Koren et al. was the first to show that the maternal gut microbiome changes during pregnancy [8]. The gut microbiome was different in pregnant women in the third trimester as compared with the first trimester, while the gut microbiome of the first trimester was not different from the general population [8]. We have recently shown that in mice the maternal gut microbiome also changes during pregnancy [9]. In view of the role of the gut microbiota in maintaining immune responses, we hypothesized that the changes in the maternal microbiome during pregnancy would be important for adapting the maternal immune response to pregnancy. We, indeed, observed different adaptations of the maternal peripheral immune response between conventional pregnant mice and germ-free pregnant mice to pregnancy. The increase in Th2 cells and Treg cells observed during pregnancy in conventional mice was not observed in germ-free pregnancy mice, while the increased activation of peripheral monocytes also was only observed in conventional pregnant mice and not in germ-free pregnant mice [9]. These data suggested that the maternal gut microbiome during pregnancy is important for some adaptations of the maternal immune response to pregnancy.

The changes in gut microbiota may induce changes in the intestinal immune system. The intestinal immune system is the largest immune system in the body [10]. This part of the immune system is exposed to gut bacteria and therefore has developed specific characteristics to protect the body from pathogens [10]. The intestinal immune system is composed of intestinal epithelial cells, the lamina propria and intestinal lymphoid tissue, such as the mesenteric lymph nodes (MLN) and Peyer’s patches (PP) [11]. The MLN and PP are important in the induction of immune tolerance and defence against pathogenic bacteria [12,13]. We have recently shown that immune changes in pregnant mice as compared to non-pregnant mice can also be observed in the intestinal immune system [14]. 

The above findings suggest that the maternal microbiome affects the intestinal immune response, which in its turn affects the maternal peripheral immune response during pregnancy. Therefore, in the present paper, we hypothesized that disturbances of the maternal microbiome would affect intestinal and peripheral maternal immune responses in pregnant mice. To disturb the maternal microbiome, pregnant mice were treated with antibiotics for 7 days. The microbiome composition was evaluated before, during and after antibiotic treatment. After the treatment, pregnant mice were sacrificed, and the peripheral and intestinal immune responses were evaluated. We found that antibiotics induced dysbiosis in the maternal gut and this was associated with aberrant intestinal and peripheral immune responses. 

## 2. Materials and Methods

### 2.1. Experimental Set-Up

Conventional C57BL/6OlaHsd mice arrived in our facility at an age of 2.5 months and were put on a standardized diet (D10012Mmi; Research Diets Inc., New Brunswick, NJ, USA) for 3 weeks before the start of experiments and remained on this diet until the end of the experiments. During their stay at our animal facility, mice were housed in individually ventilated cages and always handled in flow cabinets. After 3 weeks on the diet, vaginal smears were taken to follow the estrus cycle and female mice were put with a male mouse overnight when they were in proestrus. The next morning, when a vaginal plug was detected, that day was designated as day 0 of pregnancy. Non-pregnant mice were used as controls. Antibiotics (AB) treatment in pregnant mice started on day 9 of pregnancy and lasted for 7 days. AB were given to the mice in their drinking water (see below for details). Two days before the start of the AB treatment, 5 days after the start of AB treatment and 2 days after stopping AB treatment, feces were collected and immediately snap-frozen in liquid nitrogen. For control pregnant mice, who received regular drinking water, feces were collected and snap-frozen on the same experimental days. From non-pregnant mice, we only collected feces at day +2. Two days after stopping antibiotic treatment (i.e., day 18 of pregnancy), mice were sacrificed in sterile flow cabinets. At sacrifice, mice were anaesthetized with isoflurane and O_2_ and killed by bleeding from the heart. Blood was collected from the heart into EDTA tubes (BD-Plymouth, UK). Spleens, PP and MLNs were collected to evaluate immune responses. In pregnant mice, we counted the number of viable fetuses and resorptions and weighed individual fetuses and placentas. 

### 2.2. Animals

All experiments were approved by the Central Committee for Animal experimentation in the Netherlands (AVD1050020198488) and performed according to their guidelines. Mice (C57BL/6OlaHsd) were purchased from Envigo (Envigo, Horst, The Netherlands) and cohoused in isolated ventilated cages with 3–4 mice per cage. The light and dark cycle was a 12 h light and 12 h dark cycle and mice had ad libitum access to food and water. Mice were fed a standardized diet (D10012Mmi) (Research Diets, New Brunswick, NJ, USA). Before starting the experiments, mice were on this diet for 3 weeks. 

### 2.3. Antibiotic Treatment

Antibiotic treatment was performed according to the paper of Tochitani et al. for 7 days (from day 9 to day 16 in pregnant mice) [15]. We choose to start at day 9 of pregnancy since previous studies have shown that both the gut microbiome and the maternal immune response started to change between day 7 and 14 of pregnancy in the mouse [9]; unpublished results). On day 9 it is also able to positively verify whether a mouse is pregnant. Non-pregnant mice were treated with AB for 7 days. Mice were provided with a mix of 5 mg/mL neomycin, 5 mg/mL bacitracin and 1.25 microgram/mL pimaricin (a stock of 5 mg/mL in acetic acid was used) (all from Sigma-Aldrich) dissolved in the drinking water. These are all non-absorbable AB, which remain in the intestinal tract with limited serum concentration [15]. Control mice received regular drinking water. 

### 2.4. Isolation and Staining of T Cells and Dendritic Cells from the Spleen, MLN and PP

Isolation of cells from the spleen, MLN and PP was performed as previously described [9]. Before mechanical disruption between microscopy slides in 2 mL ice-cold RPMI containing 10% (*v/v*) decomplemented fetal calf serum (dFCS), spleens were first cut into small pieces. After disruption, splenic cells were incubated with 4 mL ice-cold ammonium chloride for 10 min to lyse the red blood cells. MLN and PP were mechanically disrupted between two microscopy slides in 2 mL ice-cold RPMI containing 10% (*v/v*) dFCS. To remove cell clumps from the splenic, MLN and PP cells, Falcon tubes with cell strainer caps (35 µm; Corning, Amsterdam, The Netherlands) were used. Then cells were counted and used for staining.

For staining of the T cell subsets, we used the antibodies shown in Appendix A. The specific antibodies to define each T cell subset are shown in Table 1. The extracellular antibodies were diluted in a volume of 25 µL in FACS buffer (Dulbecco’s phosphate buffered saline (DPBS) + 2% dFCS). The panel of intracellular antibodies was diluted in a volume of 50 µL in FACS permeabilization solution (eBioscience, Vienna, Austria). Samples were stained in 96 wells plates. Approximately 1 × 10^6^ splenic, PP or MLN cells were pipetted into the plate and washed with DPBS twice after which they were incubated in 100 µL Zombie NIR (1:1000 in DPBS; Biolegend, San Diego, CA, USA) for 30 min. After two times washing with FACS buffer, cells were resuspended in an extracellular blocking medium (20% rat serum, 78% FACS buffer, 2% anti-mouse CD16/CD32 (Biolegend)) for 10 min. After centrifugation (1800 RPM, 3 min, 4 °C), the supernatant was discarded and cells were incubated with 25 µL of extracellular antibody panel for 30 min in the dark and on ice. After two washing steps with FACS buffer, cells were incubated in FASC lysing solution (BD Biosciences, Breda, The Netherlands) for 30 min in the dark to lyse RBC and fix the cells. This was followed by two washing steps with FACS permeabilization solution (eBioscience, Vienna, Austria) and 10 min in the dark and on ice with 50 µL of intracellular blocking medium (20% rat serum, 80% FACS permeabilization solution). After centrifugation cells were resuspended in 50 µL of the intracellular antibody mix for 30 min in the dark and on ice. After washing 3 times with FACS permeabilization solution, cells were stored in 150 µL DPBS with 2% dFCS at 4 °C for a maximum of 24 h before data acquisition. 

For staining of the dendritic cells, we used the antibodies as shown in Appendix A. The specific antibodies used to define dendritic cells are shown in Table 2. The panel of antibodies was diluted in a volume of 25 µL in FACS buffer (DPBS + 2% dFCS). The samples were stained in 96 wells plates. Approximately 1 × 10^6^ splenic, PP or MLN were pipetted into the plate and washed with DPBS twice after which they were incubated in 100 µL Zombie green (1:1000 in DPBS; Biolegend) for 30 min. After 2 times washing with FACS buffer, cells were resuspended in an extracellular blocking medium (20% rat serum, 78% FACS buffer, 2% anti-mouse CD16/CD32) for 10 min on ice. After centrifugation, the supernatant was discarded, and cells were incubated with 25 µL of antibody panel for 30 min in the dark and on ice. After 2 washing steps with FACS buffer, cells were incubated in FASC lysing solution for 30 min in the dark to lyse RBC and fix the cells. After washing 3 times with FACS buffer, cells were stored in 150 µL FACS buffer at 4 °C for a maximum of 24 h before data acquisition.

### 2.5. Staining of Blood Monocytes

Maternal EDTA blood was stained for monocyte subsets and their activational status. We used the protocol previously described [9] and the antibodies described in Appendix A. The specific antibodies used to define the monocyte subsets are shown in Table 3. All antibodies were diluted in 25 µL of FACS buffer supplemented with 37.2 mg EDTA per 100 mL. Blood was diluted 1:1 with RPMI with 10% dFCS. A volume of 400 mL of diluted blood was incubated with 50 µL extracellular blocking medium (20% rat serum, 78% FACS buffer with EDTA, 2% anti-mouse CD16/32) for 10 min in the dark at room temperature (RT). After centrifugation (1800 RPM, 3 min, 4 °C), the supernatant was discarded and cells were incubated with 25 µL of antibody mix for 30 min on ice and in the dark. After 2 washing steps with FACS buffer with EDTA, cells were incubated with 1 mL FACS lysing solution for 20 min at RT. After 3 times off washing with FACS buffer with EDTA, cells were resuspended in 200 µL of FACS buffer with EDTA and stored for a maximum of 24 h at 4 °C. 

### 2.6. Flow Cytometry

We analyzed our samples with the FACSverse flow cytometer system (BD Biosciences, Franklin Lakes, NJ, USA) with the FACSvers software. Data analysis was performed using FlowJo software version 10 (FlowJo, LLC, Ashland, OR, USA). The gating strategy is described in Appendix A.

### 2.7. Microbiota Measurement

Immediately after collection, feces were snap-frozen in liquid nitrogen in screw caps and stored at −80 until measurement. The microbiota measurements were taken at Baseclear (Leiden, The Netherlands). DNA was extracted from feces using the Zymobiomics DNA mini kit (Zymo Research). DNA QC quantitation was performed with Quant-iT™ dsDNA Broad-Range Assay Kit (Invitrogen) and agarose gel electrophoresis for DNA integrity. The DNA served as a template for PCR amplification of 16S rRNA genes and subsequent analysis by next-generation sequencing using Illumina MiSeq as well as for total bacteria quantification. In short, amplicons of V3-V4 regions of 16S rRNA genes were generated by PCR with primers 341F and 785R which generated a ~630 bp amplicon [16] complemented with standard Illumina adapters. Unique Index Primers were attached to amplicons in each sample with a second PCR cycle. PCR products were purified using Agencourt© AMPure^®^ XP (Beckman Coulter Nederland B.V., Woerden, The Netherlands) and DNA concentration was measured by fluorometric analysis (Quant-it, Invitrogen, ThermoFisher Scientific, Waltham, MA, USA). Subsequently, PCR amplicons were equimolarly pooled, followed by sequencing on an Illumina MiSeq with the paired-end 300 cycles protocol and indexing. FASTQ read sequence files were generated using bcl2fastq version 2.20 (Illumina). The initial quality assessment was based on data passing the Illumina Chastity filtering. Subsequently, reads containing PhiX control signal were removed using an in-house filtering protocol. In addition, reads containing (partial) adapters were clipped (up to a minimum read length of 50 bp). The second quality assessment was based on the remaining reads using the FASTQC quality control tool version 0.11.8.

For the OTU classification, paired-end sequence reads were collapsed into so-called pseudo-reads using sequence overlap with USEARCH version 9.2 [17]. Classification of these pseudo-reads was performed based on the results of alignment with SNAP version 1.0.23 [18] against the RDP database [19] version 11.5 for bacterial organisms.

### 2.8. Universal Bacterial Quantitative Polymerase Reaction

Total bacteria levels were measured by quantitative polymerase reaction (qPCR) by using universal bacteria primers, forward primer 5′-TCCTACGGGAGGCAGCAGT-3′ and reverse primer 5′-GGACTACCAGGGTATCTAATCCTGTT-3′ [20]. Initially, DNA was diluted 10X from which 1 uL was loaded onto the plate. Then the plate was vacuum-dried (Savant SPD1010 SpeedVac concentrator, Thermofisher Scientific) and then 10 uL of FastStart Universal SYBR Green Master (Roche Diagnostics, Basel, Switzerland) master mix was added to each well. Forward and reverse primers were used at a final concentration of 300 nmol/L. qPCR was performed in a ViiATM 7 Real-Time PCR System (Applied Biosystems, Foster City, CA, USA). The cycling protocol consisted of 40 cycles of 15 s at 95 °C for denaturation, 30 s at 60 °C for primer annealing and 30 s of extension at 72 °C followed by a melting curve. Nuclease-free water, used to prepare the sample dilution, was used as a no-template control. Per plate, a 5-point standard curve built from a pool of DNA samples was used to evaluate qPCR efficiency. Only samples with Ct values <35 were considered positive for amplification. The Ct was measured, and 2^-Ct^ was used as a proxy for bacterial counts.

### 2.9. Statistics

Using GraphPad Prism (version 9), differences in weight gain between control and AB-treated pregnant and non-pregnant mice were tested using the Mann–Whitney U test; in addition, differences in placental and fetal weight as well as differences in fetal number and number of resorptions between control pregnant and AB-treated pregnant mice were tested using the Mann–Whitney U test. Data were considered significantly different if *p* < 0.05.

For evaluating the effect of pregnancy and AB on immune cells in the spleen and PP, a two-way ANOVA (TWA) was used to evaluate the effect of pregnancy, antibiotic treatment and the interaction between pregnancy and antibiotic treatment. To do so, we first tested the normality of the data using the Kolmogorov–Smirnov test. If data were not normally distributed, we log-transformed the data before performing the TWA. Since we were mainly interested in the effect of AB, post-testing was only conducted on the difference between control and AB-treated pregnant and non-pregnant mice using Sidak’s multiple comparison tests. Data were considered significantly different if *p* < 0.05.

Spearman’s correlation was evaluated between immune cells in the PP or MLN and peripheral immune cells in the spleen or blood.

For evaluating the Shannon index, the PCA, the Permanova and the similarity breakdown test (SIMPER test), we used Past4 [21]. For differences in bacterial count, Shannon index and bacterial genera between before, during and after AB treatment, the Friedman test followed by Dunn’s multiple comparisons test was used. For evaluating differences in bacterial count and Shannon index between control pregnant and pregnant AB-treated mice as well as differences between pregnant AB-treated mice and non-pregnant AB-treated mice, we used the Kruskal–Wallis test followed by the Dunn’s multiple comparisons test in which we compared all days of control pregnant mice with the respective days of AB-treated pregnant mice and all days of pregnant AB-treated mice with the respective days of AB-treated non-pregnant mice. Data were considered significantly different if *p* < 0.05.

Spearman’s correlation was also evaluated between immune cells in the spleen, PP and MLN and bacterial genera. We used bacterial genera that were different before and after AB treatment. We also added the top 10 bacterial genera from the SIMPER test. Heatmaps from the Spearman’s correlation coefficients were produced using CLUSTVIS, with cluster analysis for bacteria using Euclidean distance and Ward’s clustering [22].

## 3. Results

### 3.1. Weight Change of Pregnant and Non-Pregnant Mice following AB Treatment

Pregnant mice were weighed just before the start of AB treatment (day 0 of treatment) until the end of the experiment. Control pregnant mice gained 9.90 ± 0.46 gr (mean ± SEM) from day 9 of pregnancy to the end of the experiment, which is day 18 of pregnancy. Pregnant mice treated with AB gained 8.89 ± 0.83 (mean ± SEM) from the start of the AB treatment (day 9 of pregnancy) to the end of the experiment. The total weight gain from control pregnant mice is not significantly different as compared with AB-treated pregnant mice (Mann–Whitney U test, *t*-test, *p* > 0.05). Although the total weight gain during the experiment was not different between control pregnant and AB-treated pregnant mice, during antibiotic treatment, maternal weight gain of AB mice was significantly lower on days 4, 5, 6 and 7 after the start of the treatment (Figure 1) as compared with control pregnant mice. Non-pregnant mice were weighed just before the start of the AB treatment (day 0) and then daily until the end of the experiment. Untreated non-pregnant mice did not significantly change their weight during the experiment. Non-pregnant mice during AB treatment, however, showed a significant decrease in weight on all days during AB treatment, and their weight loss was significantly different compared with control non-pregnant mice on all days during the treatment. The weight of non-pregnant AB-treated mice returned to normal within 1 day after the end of the AB treatment.

### 3.2. Weight of Fetuses and Placentas following AB Treatment of Pregnant Mice 

The treatment of pregnant mice with AB decreased both the fetal weight as well as the placental weight as compared with fetal and placenta weight of control pregnant mice (Figure 2). We did not find differences in the total number of fetuses (control pregnant mice 6.10 ± 0.71; AB-treated pregnant mice 6.11 ± 0.64). In addition, the number of resorbed fetuses was not significantly different between control pregnant mice (0.15 ± 0.37) and pregnant mice treated with AB (0.44 ± 0.18).

### 3.3. Effect of AB Treatment on Immune Cell Populations

#### 3.3.1. Effect of AB Treatment on Th Cell Subsets in the Spleens, PP and MLN

Figure 3 shows the percentage of Th1 (T-cell-specific T-box transcription factor positive (Tbet+) Th cells), Th2 (GATA binding protein 3 positive (GATA3+) Th cells), Th17 (retinoid orphan receptor gamma t positive (RORgT+) Th cells)) and regulatory T cells (Treg; Forkhead box P3 positive (FoxP3+) Th cells) in the spleens, the PPs and the MLNs of pregnant and non-pregnant control mice and pregnant and non-pregnant mice after AB treatment.

Th1 cells were increased by AB treatment in the spleens of pregnant and non-pregnant mice (TWA, *p* < 0.05 followed by Sidak’s multiple comparison test, *p* < 0.05). Pregnancy decreased Th1 cells in the spleens (TWA, *p* < 0.05). In the PPs and the MLNs, however, no effect of AB or pregnancy was found on the Th1 cells (TWA, *p* > 0.05). We observed an effect of pregnancy and an interaction between pregnancy and AB treatment for Th2 cells in the spleen (TWA, *p* < 0.05), suggesting a differential effect of AB treatment in pregnant and non-pregnant mice. Indeed, post-testing showed a significant decrease in Th2 cells by AB treatment only in the spleens of non-pregnant mice (Sidak’s multiple comparisons tests, *p* < 0.05) not of non-pregnant mice. In both the PPs and MLNs, we also found an effect of AB treatment (TWA, *p* < 0.05), i.e., Th2 cells were decreased after AB treatment. Post-tests showed significantly decreased Th2 cells after AB treatment of non-pregnant mice compared to non-pregnant control mice in PP. 

No effect of AB treatment on Treg cells in the spleens was found (TWA, *p* > 0.05). However, in the PPs AB increased the percentage of Treg cells (TWA, *p* < 0.05). Post-testing showed that in pregnant mice, but not in non-pregnant mice, Treg cells were significantly increased (Sidak’s multiple comparison test, *p* < 0.05). In the MLNs, we found an effect of pregnancy on Tregs (TWA, *p* < 0.05). Th17 cells in the spleen were affected by pregnancy (TWA, *p* < 0.05), but not by AB (TWA, *p* > 0.05). In the PPs and MLNs AB decreased Th17 cells (TWA, *p* < 0.05); post-testing showed that in the PP, AB treatment decreased Th17 cells in pregnant and non-pregnant mice, while in the MLN, AB only tended to decrease Th17 in non-pregnant mice (Sidak’s Multiple comparison’s test, *p* = 0.0841), but not in pregnant mice. FoxP3/RoRgT double-positive cells are specific for the intestinal immune response and can only be found in the PPs and MLNs. In the PPs, FoxP3/RoRgT double-positive cells are decreased by AB treatment and by pregnancy (TWA, *p* < 0.05). Post-testing showed that in both pregnant and non-pregnant mice AB treatment decreased FoxP3/RoRgT double-positive cells (Sidak’s Multiple comparison test, *p* < 0.05). In the MLN, these cells were also decreased by AB treatment (TWA, *p* < 0.05).

#### 3.3.2. Effect of AB Treatment on Dendritic Cell Populations in the PP and MLN

Dendritic cells of the intestinal immune system are important regulators of intestinal immune responses and for the interaction of the immune system with the gut microbiota. Therefore, we evaluated the presence of dendritic cells and dendritic cell subpopulations in the PPs and MLNs (Figure 4). We found an effect of AB and pregnancy on the percentage of DC in the PP (TWA, *p* < 0.05). Post-testing showed that in non-pregnant mice, but not in pregnant mice, AB decreased the numbers of DC in the PP (Sidak’s Multiple comparison test, *p* < 0.05). We also evaluated the different subsets of DCs in the PP. The percentage of CD103+/CD11b- cells in the PP were affected by AB (TWA, *p* < 0.05), with post-testing showing that AB treatment increased the percentage of these cells only in non-pregnant mice compared to non-pregnant control mice, not in pregnant mice (Sidak’s Multiple comparison test, *p* < 0.05). The CD103-/CD11b+ dendritic cell subset in the PPs was also affected by AB treatment (TWA, *p* < 0.05); post-testing showed decreased numbers of CD103-/CD11b+ cells after AB in non-pregnant mice only (Sidak’s Multiple comparison test, *p* < 0.05). The CD103/CD11b double-positive dendritic cell population was affected by AB and pregnancy in the PPs; CD103/CD11b double-positive DCs were decreased in the PP of non-pregnant AB-treated mice compared with control non-pregnant mice (TWA, *p* < 0.05 followed by Sidak’s Multiple comparison test, *p* < 0.05).

In the MLNs the percentage of DCs was not affected by pregnancy or AB treatment (TWA, *p* > 0.05), while the percentage of CD103+CD11b- DC was affected by pregnancy and AB (TWA, *p* < 0.05) and post-testing showed an increased percentage of CD103+CD11b- DC in non-pregnant mice after AB treatment (Sidak’s multiple comparison’s test, *p* < 0.05). The CD103-CD11b+ DC cells were affected by pregnancy and there was an interaction between pregnancy and AB treatment (TWA, *p* < 0.05). This suggests a different effect of AB treatment in pregnant and non-pregnant mice. Post-testing showed that CD103-CD11b+ DC decreased only in non-pregnant AB-treated mice as compared with control mice (Sidak’s Multiple comparison test, *p* < 0.05). An interaction was also found between pregnancy and AB treatment for CD103+CD11b+ DC in the MLN (TWA, *p* < 0.05), indicating a different response upon AB treatment between pregnant and non-pregnant mice. However, we found no significant changes after post-testing. 

#### 3.3.3. Effect of AB Treatment on Blood Leukocyte Populations

We also tested the effects of AB treatment on circulating leukocytes. Although pregnancy affected peripheral leukocyte populations by decreasing the percentage of lymphocytes and increasing the percentage of granulocytes (TWA, *p* < 0.05), we found no effect of AB treatment on peripheral leukocyte percentages. (Appendix A)

##### Effect of AB Treatment on Peripheral Monocyte Subpopulations and Activation Status

Figure 5 shows the monocyte subpopulations and their activation status in non-pregnant and pregnant mice with and without AB treatment. Pregnancy, but not AB treatment, affected circulating monocyte subpopulations. Pregnancy increased the percentage of classical monocytes while decreasing the percentage of non-classical monocytes (TWA, *p* < 0.05).

Although AB treatment did not affect circulating monocyte subpopulations, it affected the activational status of monocytes. CD80 expression was increased by AB treatment on intermediate and non-classical monocytes (TWA *p* < 0.05). Post-testing showed that in both intermediate monocytes and non-classical monocytes, CD80 expression was significantly increased by AB treatment in pregnant mice, but not in non-pregnant mice (Sidak’s multiple comparison test, *p* < 0.05). The expression of MHCII was affected (decreased) by AB on classical monocytes (TWA, *p* < 0.05), while we found an effect of pregnancy and an interaction between pregnancy and antibiotic treatment for the expression of MHCII on intermediate monocytes. 

### 3.4. Gut Microbiota Composition

We determined the gut microbiota composition of pregnant and non-pregnant treated and non-treated mice by longitudinal sampling of feces before, during and after AB or control treatment. From non-pregnant control mice, we only collected feces at day +2. Unfortunately, from two non-pregnant AB-treated mice, the 16sRNA sequencing did not give good results in 1 of the samples, so these two mice were left out of the analysis of the microbiota. The bacterial count was measured using qPCR and 2^−Ct^ was used as a proxy of bacterial counts (Figure 6). Before the start of the AB treatment (day −2), bacterial counts of control pregnant (0.083 ± 0.017), pregnant AB-treated mice (0.071 ± 0.012), non-pregnant AB-treated mice (0.09 ± 0.019) and non-pregnant untreated mice (0.083 ± 0.017) did not significantly differ from each other (One way ANOVA, *p* > 0.05). Figure 7 shows bacterial counts as a percentage of day −2.

AB treatment decreased bacterial count in both pregnant and non-pregnant mice to 1–5% before treatment. In control pregnant mice, the bacterial count was significantly decreased at the end of pregnancy, i.e., day 18 of pregnancy, to about 50% of day −2 (which is day 7 of pregnancy). In untreated non-pregnant mice, the bacterial count at day +2 was similar to the bacterial count of non-pregnant AB-treated mice at day −2, i.e., before AB treatment. 

The bacterial count of pregnant AB-treated mice at day 5 and day +2 was significantly decreased as compared with the bacterial count of pregnant control mice at the same days (Kruskall–Wallis test followed by Dunn’s multiple comparison test, *p* < 0.05). There were no differences between bacterial counts at the different days of pregnant AB-treated mice and non-pregnant AB-treated mice (Kruskall–Wallis test followed by Dunn’s multiple comparison test, *p* > 0.05).

Also, the alpha diversity, as measured by the Shannon index is decreased after AB treatment (Figure 7). 

The Shannon index of control pregnant mice did not change during pregnancy (Friedman, *p* > 0.05). However, in AB-treated pregnant mice, the Shannon index decreased significantly during AB treatment; 2 days after AB treatment, the Shannon index was not significantly different from the Shannon index of the mice before the start of the AB treatment. In non-pregnant mice, the Shannon index significantly decreased during AB treatment and remained significantly decreased 2 days after AB treatment. In untreated non-pregnant mice, the Shannon index at day +2 was similar to the Shannon index of non-pregnant AB-treated mice at day −2, i.e., before AB treatment (results not shown). There were no differences in Shannon index between the different groups (Kruskall–Wallis test followed by Dunn’s multiple comparison test, *p* > 0.05)

Figure 8 shows the changes in abundance of the different bacterial phyla in control pregnant mice and in pregnant and non-pregnant mice before, during and after AB treatment.

In pregnant control mice, we found an increase in Firmicutes and a decrease in Bacteroidetes at day +2 as compared with day −2 (Friedman’s test followed by Dunn’s multiple comparisons test, *p* < 0.05). This resulted in a significantly increased Firmicutes/Bacteroidetes ratio at day +2 vs. day −2 (Friedman’s test followed by Dunn’s multiple comparisons test, *p* < 0.05; results not shown). AB treatment during pregnancy induced a variable change in the abundance of the different fecal bacterial phyla. AB treatment hugely increased the abundance of Bacteroidetes in four out of nine pregnant mice during AB treatment (day 5) and in three out of nine mice after AB treatment (day +2). After AB treatment, one mouse showed a hugely increased abundance of Proteobacteria. AB treatment in non-pregnant mice induced a more consistent change in fecal bacterial phyla; it significantly increased the abundance of Bacteroidetes in mice both during and after AB treatment (Friedman’s test followed by Dunn’s multiple comparisons test, *p* < 0.05). In untreated non-pregnant mice, the phyla abundance at day +2 was similar to the phyla abundance of non-pregnant AB-treated mice at day −2, i.e., before AB treatment (results not shown).

To further investigate bacterial differences between the groups, we analyzed the difference in bacterial genera before, during and after AB treatment in pregnant and non-pregnant mice. We performed PCA analysis (Figure 9) on the level of genera for pregnant control mice (left) pregnant AB-treated (middle) as well as for non-pregnant AB-treated mice (right). See also Appendix A.

In pregnant mice, we found clear differences in the microbiota genera before, during and after AB treatment. The analysis showed that the principal component (PC1), represented by AB treatment, explained about 70% of the variation of the genera between the groups. PC2, representing the time after AB treatment, explained about 30 % of the variation. Both day 5 as well as day +2 were significantly different from day −2 (Permanova, *p* < 0.05). In addition, for non-pregnant mice, we found clear effects of AB treatment on the microbial genera. PC1, represented by AB treatment, explained about 90% of the variation, while PC2, represented by time after treatment, explained about 10% of the variation. Both day 5 as well as day +2 were significantly different from day −2 (Permanova, *p* < 0.05). The Permanova also showed that the microbiota genera of control pregnant mice at day 5 and day +2 were significantly different from the microbiota genera of AB-treated pregnant mice at the same days (Permanova, *p* < 0.05). There was also a difference in the microbiota genera between pregnant and non-pregnant AB-treated mice; at day +2 the microbiota genera of pregnant AB-treated mice was significantly different as compared with the microbiota genera from non-pregnant AB-treated mice at the same day (Permanova, *p* < 0.05).

The top five bacterial genera in pregnant mice explaining the changes in microbiota after AB treatment are the genus *Alistipes*, genus *Barnesiella*, genus *Faecalibaculum*, genus *Lachnoclostridium* and genus *Escherichia* (SIMPER test (Appendix A); this table also gives a complete overview of the significant changes in genera during and after treatment as compared with before treatment). The top five bacterial genera in non-pregnant mice were the genus *Alistipes*, genus *Bacteroides*, genus *Barnesiella*, genus *Faecalibaculum* and genus *Akkermansia* (SIMPER test Appendix A).

Figure 10 shows examples of changes in the abundance of bacterial genera during and after AB treatment as compared with before treatment in pregnant mice. We show the top five genera of the SIMPER test (as indicated above) supplemented with the top five bacterial genera of the SIMPER test that were significantly changed at day +2 compared with day −2. As can be seen from Figure 10, the response of the gut bacterial genera to AB treatment was variable. Four mice showed a huge increase in the genus *Alistipes* at day 5 (to above 90% abundance), two mice showed an increase in the genus *Escherichia* and two mice showed an increase in the genus *Faecalibaculum* (*Faecalibaculum rodentium*). Concerning the genus *Alistipes*, one mouse showed a huge increase in *Alistipes timonensis*, while three mice showed a huge increase in *Alistipes finegoldii*. This increase in specific bacterial genera was amongst others associated with a decrease in the genus *Barnesiella* (*Barnesiella intestinihominis, Barnesiella viscericola and unclassified Barnesiella*), genus *Lachnoclostridium*, unclassified genus *Bacteroidales*, genus *Alloprevotella*, genus *Odoribacter* and genus *Prevotella*. (See also Appendix A). Examples of changes in the abundance of bacterial genera in non-pregnant mice are shown in Appendix A.

To get insight into the relationship between the gut microbiota and the immune response in AB-treated mice, we correlated individual microbiota abundance data with individual immune cell data of the same AB-treated mice. Pearson’s correlation coefficients for peripheral immune cells and PP and the MLN of pregnant mice are shown in a heatmap (Figure 11). In pregnant AB-treated mice, immune cells in the PP and MLN correlated with various bacterial genera. In the PP, various bacterial genera, including genus *Akkermansia* or genus *Faecalibaculum*, correlated negatively with the CD103+CD11b+ dendritic cells. These genera also correlated positively with Treg cells in the PP. In the PP, the genus *Escherichia* correlated positively with FoxP3/RoRgT double-positive Th cells and with CD103+CD11b- dendritic cells. In the MLN, the genus *Lachnoclostrium* and the genus *Flintibacter* correlated positively with CD103+CD11b+ dendritic cells. Other bacterial genera, i.e., genus *Anaerostipes*, genus *Lactobacillus* and genus *Akkermansia*, correlated negatively with Th2 and Th17 cells in the MLN. Genus *Alistipes* was positively correlated with Th1 cells. For peripheral immune cells, many bacterial genera correlated positively with CD80 positive intermediate and non-classical monocytes, while also various genera correlated negatively with Th2 and Th17 cells. Genus *Alistipes* and genus *Oribacterium* correlated positively with Th1 cells. Heatmaps of Pearson’s correlation coefficients for peripheral, PP and MLN immune cells with microbial abundance data of non-pregnant mice are shown in the Appendix A.

## 4. Discussion

In the present study, we showed that treatment with AB during pregnancy in mice not only affected the maternal fecal microbiome but also affected the maternal immune response. Since the AB used were non-absorbable, we do not expect direct effects of the AB on the maternal immune response, suggesting that the disturbances in the maternal microbiome induced by the AB caused the aberrant maternal immune responses. We indeed found various correlations between fecal bacterial genera and maternal immune cells in the MLN and PP as well as in the spleen. The present data confirm our previous study in which we showed that adaptations in maternal immune responses are (partly) caused by the maternal microbiome [9].

During pregnancy, immune responses have to adapt to the presence of the semi-allogeneic fetus. The present data confirm these adaptations in peripheral immune responses since we found a decreased Th1 and Th17 cells in pregnant compared with non-pregnant mice, together with changes in monocyte subsets and an increased activational status of monocytes in pregnant compared with non-pregnant mice. The present data for the first time showed changes in immune cells in the PP and the MLN during pregnancy: we observed a decreased percentage of Th17 cells together with an increased percentage of Treg cells in the MLN of pregnant mice as compared with non-pregnant mice. In the PP, we observed a decreased percentage of FoxP3/RoRgT positive Th cells during pregnancy. This study also for the first time showed changes in dendritic cells in the PPs and MLNs during pregnancy; dendritic cell number and the CD103+CD11b+ dendritic cells were decreased during pregnancy in the PP, while in the MLNs CD103+CD11b- are decreased and CD103-CD11b+ dendritic cells are increased during pregnancy. Dendritic cells in the intestinal immune system are important regulators of immune responses in the intestinal immune system and are known to be affected by the microbiome [10]. The intestinal immune response in its turn can affect peripheral immune responses; therefore, we hypothesize that the intestinal dendritic cells may play a role in the adaptations of the peripheral immune response during pregnancy.

The effects of the AB treatment on the immune cells were mainly similar between pregnant and non-pregnant mice, although the size of the effect differed for some cell types. We observed increased Th1 cells and decreased Th2 cells in the spleens of both pregnant and non-pregnant AB-treated mice. We also found changes in the activational status of circulating monocytes in AB vs. control pregnant and non-pregnant mice; MHCII expression on classical monocytes after AB treatment was decreased, while CD80 expression on non-classical and intermediate monocytes was increased. Decreased monocyte MHCII expression indicates a decreased ability of monocytes to present antigens and thus disturbed T cell stimulation [23]. Increased expression of CD80 on monocytes indicates the activation of the monocytes [24]. In a recent study, Benner et al. also observed the effects of AB on the maternal immune response in pregnant mice [25]. They observed effects in the peripheral immune cells and placenta, but not in the MLNs. Differences between the present study and the study of Brenner can be explained by the fact that they used a different cocktail of AB and also a different timing of AB treatment.

We observed effects of AB treatment on immune cells in the PP and MLN. After AB treatment, we found increased Treg cells and decreased Th2, Th17 and FoXP3/RoRgT double-positive Th in the PP. Th17 cells in the PP are mainly induced by segmented filamentous bacteria (SFB), that form colonies on small intestine epithelial cells [26]. Our data suggested that these bacteria are decreased by the AB treatment. Since SFB mainly reside in the small intestines, these bacteria were not found in the mice feces we collected. The FoXP3/RoRgT double-positive Th cells are known to be largely decreased in germ-free mice and thus induced by the microbiota [27]. The large decrease in bacterial number in our study after AB treatment may thus explain the decreased numbers of FoXP3/RoRgT double-positive Th cells in the PPs. It has recently been shown that these FoXP3/RoRgT double-positive cells regulate Th2 responses, since the lack of these cells in mice, increased Th2 cells in the intestinal immune system [27]. However, although decreased FoXP3/RoRgT double-positive Th cells were found in the PP of AB-treated mice in our study, this was associated with decreased Th2 cells in the PP of these mice. Apparently, in our study, other mechanisms have a stronger influence than FoXP3/RoRgT double-positive Th cells and decreased Th2 cells in the PP of AB-treated mice.

After AB treatment of pregnant mice, various bacterial genera correlated with CD103+CD11b+ dendritic cells in the PP or MLN. The genera *Odoribacter*, *Lachnoclostridium*, *Flintibacter* and *Acetatifactor* positively correlated with the MLN CD103+CD11b+ dendritic cells. In the PP, the genera unclassified *Bacteroidales*, *Lactobacillus*, *Akkermansia*, *Faecalibaculum*, unclassified *Erysipelotrichales*, *Odoribacter* and *Alloprevotella* correlated negatively with this dendritic cell population. This dendritic cell population is unique to the intestinal immune system [10] and drives Th17 differentiation in the intestinal immune response [28]. The Th17 cytokines support the integrity of the gut barrier [29]. We found slightly decreased CD103+CD11b+ dendritic cells in the PPs and MLNs of pregnant AB-treated mice, which was associated with decreased Th17 cells in the PP and MLN. These data suggest that CD103+CD11b+ in the PPs and MLNs are affected by various bacterial genera after AB treatment during pregnancy and that these cells decreased Th17 cells in the PPs and MLNs. Since Th17 cells support the intestinal barrier, the decreased Th17 cells after AB treatment may be associated with increased intestinal permeability. Indeed, AB treatment has been shown to affect intestinal barrier function [30]. Future studies should address intestinal permeability in pregnant AB-treated mice and whether an increased intestinal permeability is associated with changes in the maternal immune response.

Peripheral immune responses were also affected by the AB treatment in mice. This may be due to the changes in intestinal immune responses. We found several correlations between immune cells in the PPs and MLNs and peripheral immune cells in the spleen or blood (Appendix A). Peripheral immune responses can also be affected by the production of microbial products, secreted into the peripheral circulation. Short-chain fatty acids (SCFA), such as butyrate, propionate and acetate, are microbial products secreted into the circulation and are well-known for their effects on immune cells [31]. As dysbiosis is associated with changes in the production of SCFA [31], in our study different plasma levels of SCFA after AB treatment may have affected the peripheral immune cells. Unfortunately, we were not able to collect enough plasma to measure the SCFA in the plasma.

The AB treatment induced dysbiosis in both pregnant and non-pregnant mice: the AB induced a decrease in bacterial count and Shannon diversity during and 2 days after AB treatment, which is similar in both pregnancy and non-pregnant mice. Interestingly, in pregnant mice the response to AB was variable, i.e., different bacterial genera became dominant in different mice. The reason for this is unclear since we used inbred mice that were kept on a standard diet (research diets) and under standard circumstances. It was also not a cage effect, since for instance the four mice that increased the genus *Alistipes* were kept in three different cages. The response of non-pregnant mice was more consistent in that all mice increased the genus *Alistipes*, although in some mice to a very low extent. Still, also non-pregnant mice responded variably since only a few mice increased the genus *Bacteroides* and only one mouse increased the genus *Feacalibaculum*. The reason for these variable responses is unclear and needs further investigation.

Our data indicate that AB treatment during mouse pregnancy affects the maternal innate and adaptive immune response, both peripherally as well as in the intestines. The present study was set up to test the hypothesis that disturbances of the maternal microbiome affected maternal immune responses. Our experimental setup of 7 days of AB treatment, i.e., 1/3 of mouse pregnancy, does not have a direct clinical equivalent. However, our findings do have clinical relevance, since our data indicate that disturbances of the maternal microbiome not only affected maternal immune responses, but also pregnancy outcomes in mice, i.e., decreased fetal and placenta weight. Since the adaptation of the immune response to pregnancy are similar in mouse and human pregnancy (decreased Th1 and Th17 and increased Th2 and Treg [9,32]), AB treatment in human pregnancy may also result in a disturbance in the maternal microbiome and thus disturbance of the maternal immune responses. This may affect pregnancy outcomes, since it is well-known that in human pregnancy, aberrant maternal immune responses are associated with various pregnancy complications, such as fetal growth restriction or preeclampsia [33]. In our study, AB treatment induced an increase in the peripheral Th1 response during pregnancy. An increase in the Th1 immune response is detrimental to pregnancy and is associated with pregnancy complications such as preeclampsia, fetal growth restriction, abortion or preterm labour [34,35,36,37]. We thus hypothesize that the changes in the maternal immune response after AB treatment induced decreased fetal and placental weight in pregnant mice. In addition, in humans, AB treatment affected fetal weight [38,39,40]. The mechanism for the AB effect in human pregnancy was unknown until now. Our data, however, show that disturbances of the maternal microbiome inducing aberrant maternal immune responses may be the underlying mechanism.

*Concluding remarks:* We found that AB treatment in pregnant mice induced gut dysbiosis. Since the AB were non-absorbable [15], we do not expect a direct effect of the AB on the immune response. We thus suggest that the gut dysbiosis affected the intestinal immune response, which in turn affected the peripheral immune response. However, changes in the peripheral immune response may also be induced by changes in microbial products, such as SCFA, as a result of the dysbiosis. Changes in the maternal immune response may induce changes in placental development and therefore decreased placental and fetal weight. As various pregnancy complications, such as preeclampsia or preterm birth, are induced by aberrant maternal immune responses [33], the question arises whether these pregnancy complications are also associated with dysbiosis of the maternal gut. Recent data showed that this may indeed be the case; preeclampsia, for instance, is associated with dysbiosis of the maternal microbiome [41]. We therefore suggest that pregnancy complications induced by aberrant maternal immune responses may be caused by maternal gut dysbiosis. Therefore, correction of dysbiosis, by for instance pre-, pro- or postbiotics, may improve or prevent such complications. Future studies are necessary to test this hypothesis.

## Figures and Tables

**Figure 1 nutrients-15-02723-f001:**
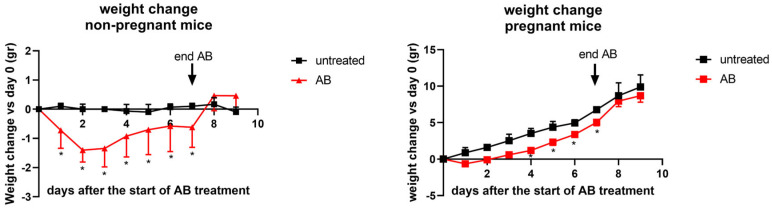
Weight changes of untreated and AB-treated pregnant (left graph) and non-pregnant (right graph) mice. For pregnant mice: day 0 (the start of the experiment) was day 9 of pregnancy; day 9 was day 18 of pregnancy, i.e., day of sacrifice. Pregnant AB-treated mice: n = 9; untreated pregnant mice: n = 9; non-pregnant AB-treated mice: n = 10; untreated non-pregnant mice: n = 11. *: AB-treated mice vs. control mice on the same day of treatment. Mann–Whitney U test, *p* < 0.05.

**Figure 2 nutrients-15-02723-f002:**
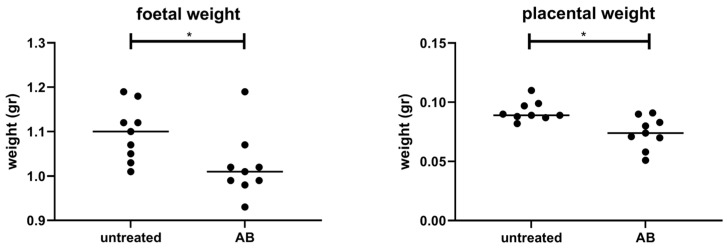
Mean fetal and placental weight of all pups and placentas per dam after in untreated or AB-treated pregnant mice. Pregnant AB-treated mice: n = 9; untreated pregnant mice: n = 9. *: Mann–Whitney U test, *p* < 0.05.

**Figure 3 nutrients-15-02723-f003:**
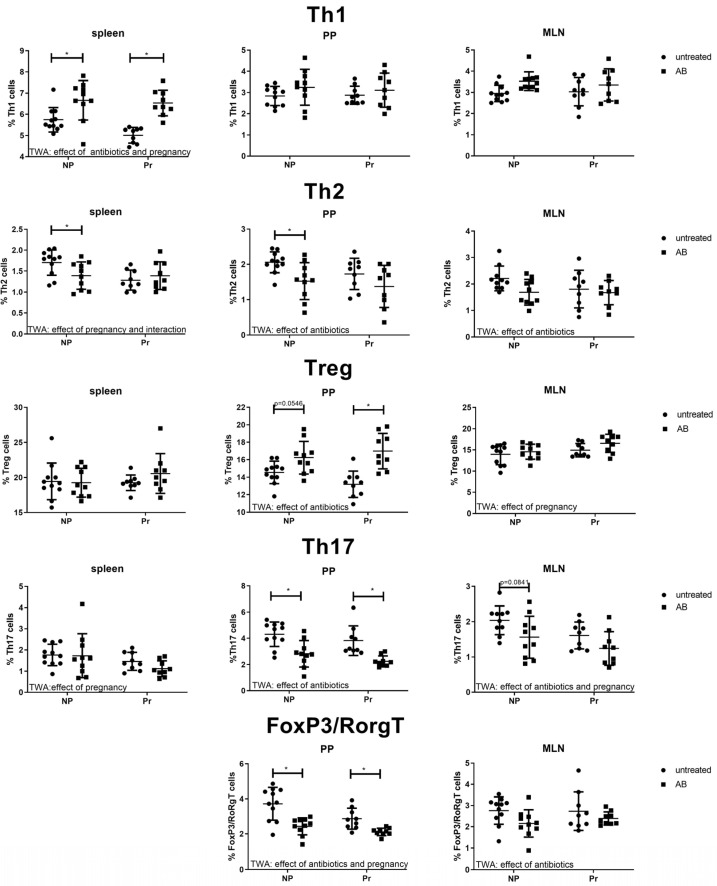
The T cell subsets Th1 (Tbet+ Th cells), Th2 (GATA3+ Th cells), Treg (Regulatory T cells (FoxP3+ Th cells)), Th17 (RoRgT+ Th cells) and FoxP3/RoRgT double-positive Th cells in the spleens, Peyer’s patches (PP) and mesenteric lymph nodes (MLN) of pregnant and non-pregnant antibiotic-treated (AB) and control mice. Pregnant AB-treated mice: n = 9; untreated pregnant mice: n = 9; non-pregnant AB-treated mice: n = 10; untreated non-pregnant mice: n = 11. *: Two-way ANOVA was performed for all subsets in the spleens, PP and MLN followed by Sidak’s multiple comparison test, *p* < 0.05.

**Figure 4 nutrients-15-02723-f004:**
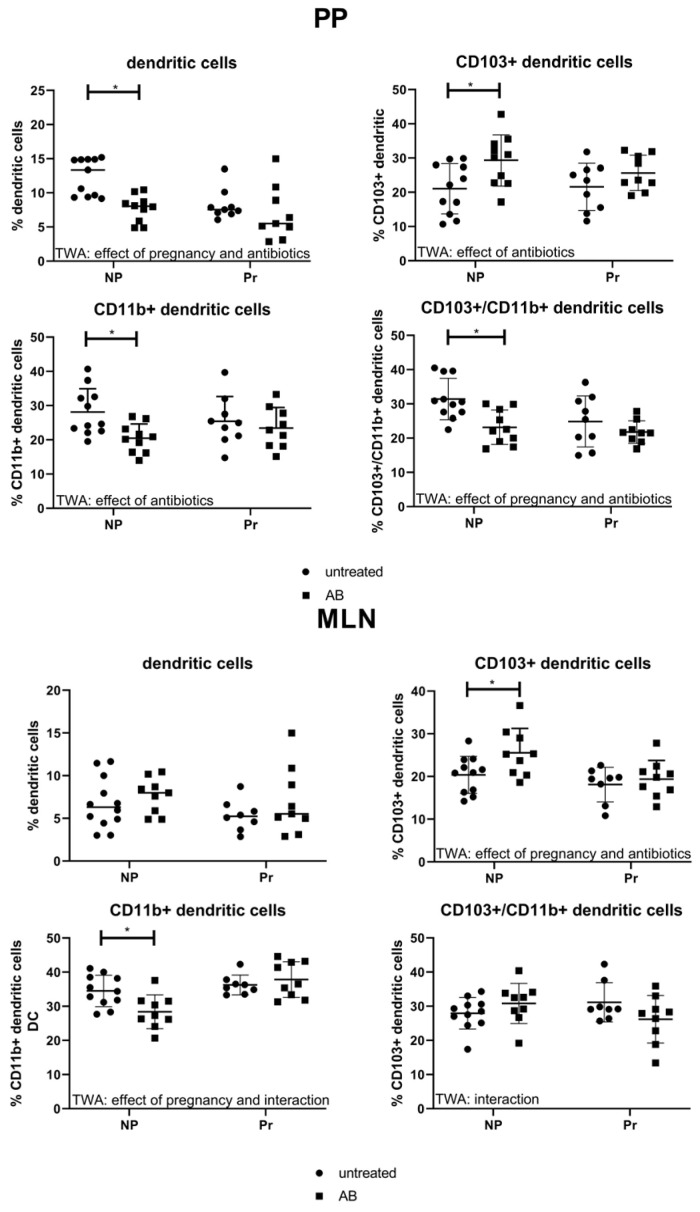
Percentage of dendritic cells and dendritic cell populations in the PP (left) and MLN (right) of pregnant and non-pregnant control and AB-treated mice. Pregnant AB-treated mice: n = 9; untreated pregnant mice: n = 9 (n = 8 for MLN, from 1 mouse, the MLN was not isolated); non-pregnant AB-treated mice: n = 10 (n = 9 for MLN, from 1 mouse the MLN was not isolated); untreated non-pregnant mice: n = 11. *: Two-way ANOVA was performed for all subsets in the PP and MLN, followed by Sidak’s multiple comparison test, *p* < 0.05.

**Figure 5 nutrients-15-02723-f005:**
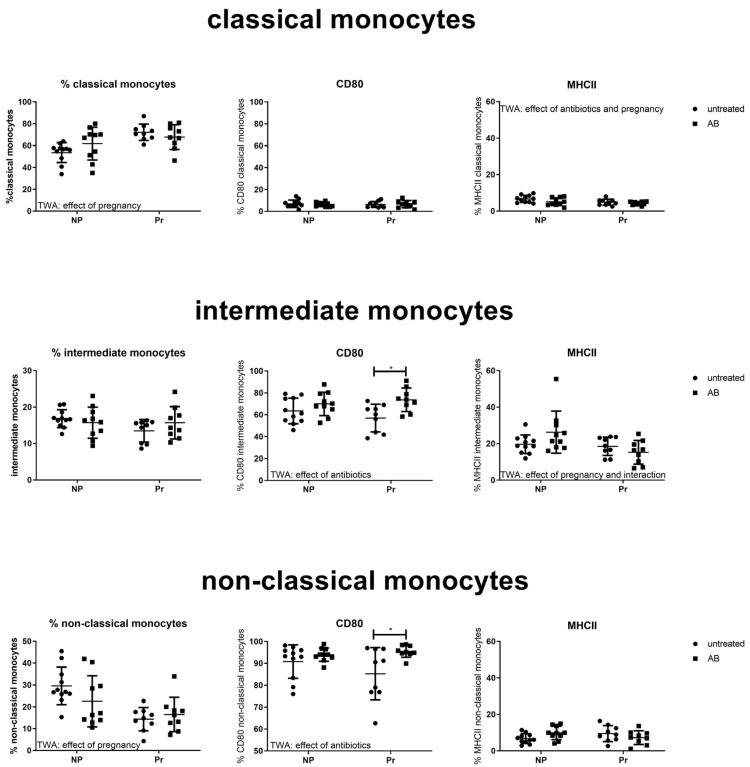
Blood monocyte subsets and activational status, as measured by expression of CD80 and MHCII, in pregnant and non-pregnant control and AB-treated mice. Pregnant AB-treated mice: n = 9; untreated pregnant mice: n = 9; non-pregnant AB-treated mice: n = 10; untreated non-pregnant mice: n = 11. *: Two-way ANOVA was performed for all subsets, followed by Sidak’s multiple comparison test, *p* < 0.05.

**Figure 6 nutrients-15-02723-f006:**
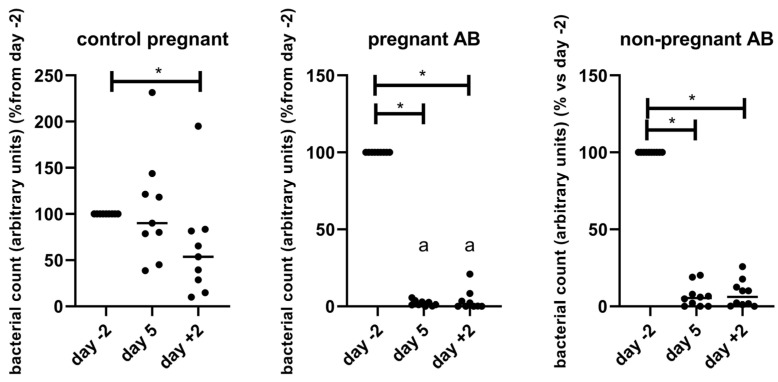
Bacterial counts, as a percentage of day −2, i.e., before treatment, in control pregnant mice and in pregnant and non-pregnant AB-treated mice. Bacterial counts were measured using qPCR and fold change of delta Ct was used as a proxy of bacterial counts. Pregnant AB-treated mice: n = 9; untreated pregnant mice: n = 9; non-pregnant AB-treated mice: n = 8. *: Friedman followed by Dunn’s multiple comparison test, *p* < 0.05. a: significantly different from control pregnant mice on the same day. Kruskall–Wallis test followed by Dunn’s multiple comparison test, *p* < 0.05.

**Figure 7 nutrients-15-02723-f007:**
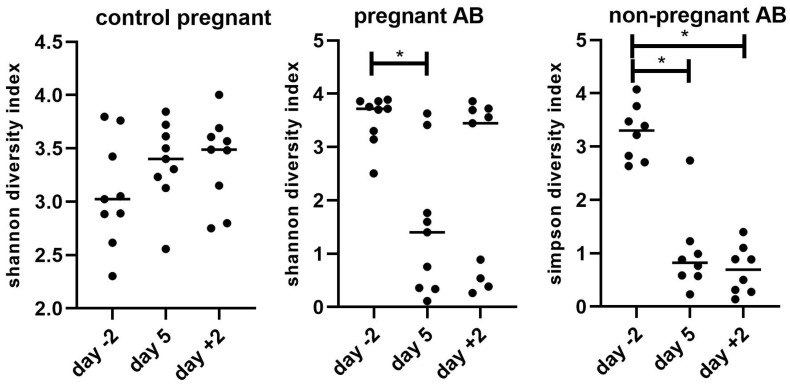
Shannon diversity index of control pregnant mice and of pregnant and non-pregnant AB-treated mice, before, during and after AB treatment. Pregnant AB-treated mice: n = 9; untreated pregnant mice: n = 9; non-pregnant AB-treated mice: n = 8. *: Friedman followed by Dunn’s multiple comparison test, *p* < 0.05.

**Figure 8 nutrients-15-02723-f008:**
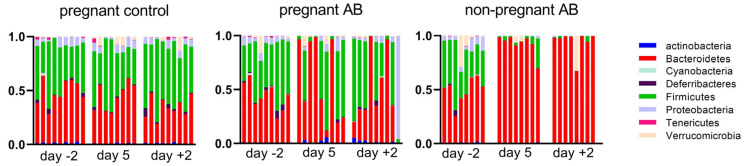
Bacterial phyla in control pregnant mice and in AB-treated pregnant and non-pregnant mice before (day −2), during (day 5) and after treatment (day +2). Pregnant AB-treated mice: n = 9; untreated pregnant mice: n = 9; non-pregnant AB-treated mice: n = 8.

**Figure 9 nutrients-15-02723-f009:**
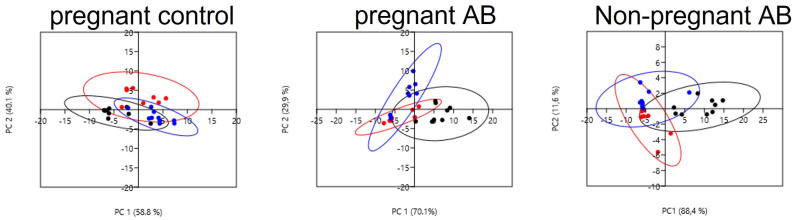
PCA plots of bacterial genera in control pregnant mice and before, during and after AB treatment in pregnant (**left**) and non-pregnant mice (**right**). (Black dots: day −2; red dots: day 5; blue dots: day +2). Pregnant AB-treated mice: n = 9; untreated pregnant mice: n = 9; non-pregnant AB-treated mice: n = 8.

**Figure 10 nutrients-15-02723-f010:**
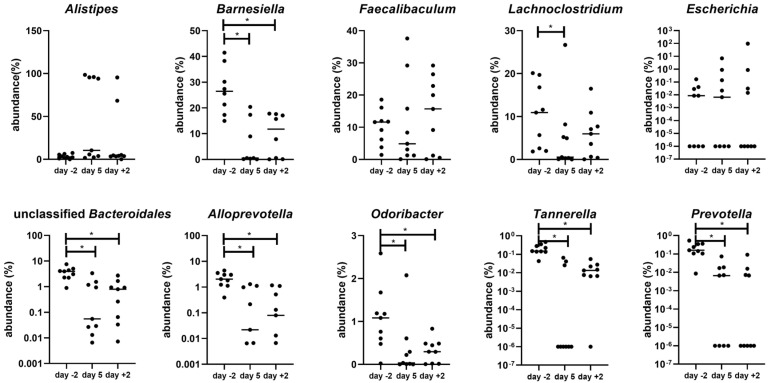
Various microbial genera before (day −2) and during (day 5) and after (day +2) AB treatment of pregnant mice. (n = 9). *: Friedman followed by Dunn’s multiple comparison test, *p* < 0.05.

**Figure 11 nutrients-15-02723-f011:**
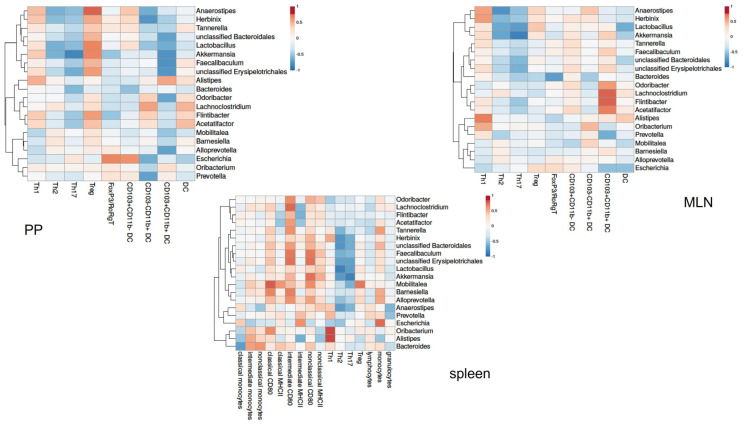
Correlation between immune cell populations in the PP (top left heatmap) and MLN (top right heatmap) and peripheral cells (lower heatmap) in pregnant mice. The heatmaps show Spearman’s correlations coefficients after individual correlation of PPs, MLN or peripheral immune cells (on the *x*-axis) with bacterial genera that were significantly different between day −2 and day +2. Pregnant AB-treated mice: n = 9; untreated pregnant mice: n = 9.

**Table 1 nutrients-15-02723-t001:** Antibodies defining T cell subsets.

T Cell Subset	Antibodies
Th1	CD3+CD4+Tbet+
Th2	CD3+CD4+Gata3+
Th17	CD3+CD4+RoRgT+
Treg	CD3+CD4+FoxP3+
FoxP3+/RoRgT+	CD3+CD4+FoxP3+RoRgT+

**Table 2 nutrients-15-02723-t002:** Antibodies defining dendritic cell subsets.

Dendritic Cell Subsets	Antibodies
Dendritic cell population	MHCII+CD64-CD11c+
CD103+ dendritic cells	MHCII+CD64-CD11c+CD103+
CD11b+ dendritic cells	MHCII+CD64-CD11c+CD11b+
CD103+CD11b+ dendritic cells	MHCII+CD64-CD11c+CD103+CD11b+

**Table 3 nutrients-15-02723-t003:** Antibodies defining monocyte subsets.

Monocyte Subsets	Antibodies
Classical monocytes	CD11b+Ly6G-Ly6C++CD43dim
Intermediate monocytes	CD11b+Ly6G-Ly6C+CD43dim
Non-classical monocytes	CD11b+Ly6G-Ly6CdimCD43+

## Data Availability

The raw data supporting the conclusions of this manuscript will be made available by the authors upon request.

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
