# Peer review of "The Effect of Antibiotics Treatment on the Maternal Immune Response and Gut Microbiome in Pregnant and Non-Pregnant Mice"

_nutrients, 2023, doi:10.3390/nu15122723_

Round 1

Reviewer 1 Report

In a factorial design authors provided antibiotic water to pregnant and non-pregnant mice for one week starting at day 9 of pregnancy, collected blood and tissue samples, and made various measurements at different time points. Authors may wish to  address the following specific comments:

1. Methods of statistics were described in detail between lines 221 and 248. It was not clear to this reviewer as whether interaction between the two main effects (pregnancy and AB) was part of the model. If not, why not? For most of the data presented in the figures, no interaction effect was declared or discussed. The interaction term in the two way variance analysis, which defines differential AB effects on pregnant and non-pregnant mice, is very important and very much relevant to the study. If there is misinterpretation from this reviewer over this issue, authors should make things clear so that readers will not have misinterpretations. 

2. Disclose very clearly in the material and method section what specific markers (antibodies) were used by authors to define each cell population (eg Th1, Th2, Th17, Treg). Label clearly in the FACS plots the exact antibody used to define various cell populations (showing the dye is insufficient).

3. Make an effort to reorganize data to reduce the number of figures. The three Tables could all be moved to supplemental material.

4. Show the control data (non-pregnant non AB treatment) in Figures 7-10, in addition to data currently shown for the other three groups. 

5. Data presented in Figure 9 (Lin 379-395) seemed to show a clear difference between pregnant and non-pregnant mice in response to AB treatment: pregnant mice had better maintenance of Firmicutes-Bacteroidetes balance at day 5 and had quick restoration toward normal  Firmicutes-Bacteroidetes balance at day +2. This observation would imply that pregnant mice are more tolerate toward AB treatment than non-pregnant mice.    

6. Be consistent in data presentation and legend description. Data presented in the figures were inconsistent with description in the figure legends in terms of the number of observations used for analysis. For example, in figure 9 for the non-pregnant AB group, legend specified N=10 (line 503) yet in the actual figure only 8 observations were presented. What happened to the other two observations? This happened in figure panels as well in which numbers of data dots do not correspond with N number specified in the legend. Authors need to make a serious effort to examine all figures to make sure that statistical analysis and conclusion were based on all data, not part of the data. Having missing values is understandable but explanation should be provided in each case.

The manuscript reads OK to me.

Reviewer 2 Report

This is an innovative, well-structured and well-founded article on the effects of antibiotic therapy and on intestinal microbiota dysregulation and maintenance of immune mechanisms, as well as the regulatory role promoted by the gestational stage.

The article presents a structured and hierarchical introduction and justification, consistent and applicable methodology for the intended objectives of the research. As for the results, they deal with a rich exposition of adequately described data that is easy for the reader to understand, in addition to a rich statistical evaluation that underlies the referred results; followed by plausible discussion that supported the findings of the work.

Author Response

We thank the reviewer for his/her positive comments!

Reviewer 3 Report

The author have set out to demonstrate that "inducing gut dysbiosis during pregnancy altered the maternal immune response in pregnant rats."  Because the errors in statistical interpretation described in the "Methods" section were not repeated anywhere else in the paper, as the data and their interpretation have been presented, the authors have seemingly achieved their goal.  However:

lines 227 and 234-235 and 242-243: "A trend was defined when the p-value was between 0.05 and 0.1." - NO.  This is a common misuse of the word "trend."  You seem to understand this by not using the word "trend" anywhere except in the "Methods" section of your paper.  Please look this up and correct your approach.  Invalid approaches in the statistical analysis of even part of the data reduce the reliability of the entire set of findings.

What does the "*" mean in Tables S1 and s@?

Round 2

Reviewer 1 Report

The revised manuscript showed some improvements.  Still, problems exists in data analysis/presentation as pointed out in previous comment 7. Notably authors changed N values for Figures 7 &  8 (previous figures 8 & 9) in legends.  As pointed out earlier, authors should make an effort to examine data presented in all figure panels to make sure that data in display  correspond to the N values specified in figure legends. Some inconsistence still exist in the revised figures.  

Author Response

We thank the reviewer for this comment. We agree that some inconsistencies still exist in the revised paper and we are sorry for this. 

Unfortunately, from 2 non-pregnant AB treated mice, the 16sRNA sequencing did not give good results in 1 of the samples, so these 2 mice were left out of the analysis of the microbiota, so in this group n=8 for the microbiota data. This was not well implemented in all figures and legends. For the immune cell data, all mice were included, resulting in an n=10 for this group in the immune cells data.

However, for dendritic cell subsets in the MLNs (figure 4), for 1 untreated pregnant mouse and 1 non-pregnant AB-treated mouse, MLNs were not collected. This has been indicated in the legend of figure 4 (lines 489-490 of the revised version).

At lines 364-366 in the revised version of the manuscript, we have now included the following sentence: "Unfortunately, from 2 non-pregnant AB treated mice, the 16sRNA sequencing did not give good results in 1 samples, so these 2 mice were left out of the analysis of the microbiota." 

Reviewer 3 Report

Nice work addressing my previous issues.  Thank you.

Author Response

Thank you!